# An Asymptotic Test for Bimodality Using The Kullback–Leibler Divergence

**Javier E. Contreras-Reyes** 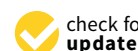

Departamento de Estadística, Facultad de Ciencias, Universidad del Bío-Bío, Concepción 4030000, Chile; jcontreras@ubiobio.cl

**Abstract:** Detecting bimodality of a frequency distribution is of considerable interest in several fields. Classical inferential methods for detecting bimodality focused in third and fourth moments through the kurtosis measure. Nonparametric approach-based asymptotic tests (DIPtest) for comparing the empirical distribution function with a unimodal one are also available. The latter point drives this paper, by considering a parametric approach using the bimodal skew-symmetric normal distribution. This general class captures bimodality, asymmetry and excess of kurtosis in data sets. The Kullback–Leibler divergence is considered to obtain the statistic's test. Some comparisons with DIPtest, simulations, and the study of sea surface temperature data illustrate the usefulness of proposed methodology.

**Keywords:** bimodality; bimodal skew-symmetric normal distribution; Kullback–Leibler divergence; Shannon entropy; sea surface temperature

## 1. Introduction

The bimodality of a frequency distribution is crucial in several fields. For example, [1] analyzed bimodality-generating mechanisms for age-size plant population data sets. Ashman et al. [2] discussed the presence of bimodality in globular cluster metallicity distributions, velocity distributions of galaxies in clusters, and burst durations of gamma-ray sources. Hosenfeld et al. [3] detected bimodality in samples of elementary schoolchildren's reasoning performance. Bao et al. [4] applied the minimum relative entropy method for bimodal distribution to remanufacturing system data. Freeman and Dale [5] assessed bimodality to detect the presence of dual cognitive processes, and Shalek et al. [6] found bimodal variation in immune cells using ribonucleic acid (RNA) fluorescence.

In the literature, exist several measures of a frequency distribution's bimodality. Classical inferential methods for detecting bimodality focused on third and fourth moments through kurtosis measure [1]. Darlington [7] and Hildebrand [8] claimed that kurtosis is more a measure of unimodality versus bimodality than a measure of peakedness versus flatness. Hartigan and Hartigan [9] considered an asymptotic test to compare the empirical distribution function with a unimodal one. This paper is motivated by the latter, but considers a parametric approach. Specifically, we considered a generalized class of distributions that involves bimodal behavior in empirical distribution. This class is called the bimodal skew-symmetric normal (BSSN) distribution [10], and includes the particular case of bimodal normal distribution of [11]. Thus, with BSSN distribution it is possible to capture asymmetric and platykurtic/leptokurtic (excess negative/positive kurtosis) in data sets, in addition to bimodality. Besides, entropic measures are useful to obtain the statistic's test if some regularity conditions of the probability distribution function are accomplished [12]. We considered the Shannon entropy [13], the Kullback–Leibler [14] divergence, and the BSSN maximum likelihood estimators to provide an asymptotic test for bimodality.

This paper is organized as follows: some properties and inferential aspects of BSSN distribution are presented in Section 2. In Section 3, we provide the computation and description of information

theoretic measures related to BSSN distribution and then develop a hypothesis test about significance of bimodality parameter together with a simulation study (Section 4). In Section 5, real data of sea surface temperature collected off northern Chile illustrate the usefulness of the developed methodology. Discussion concludes the paper in Section 6.

## 2. Bimodal Skew-Symmetric Normal Distribution

**Definition 1.** *Let X be a continuous random variable defined at $\mathbb{R}$, so we say X is bimodal skew-symmetric normal (BSSN), distributed and denoted as $X \sim BSSN(\mu, \sigma^2, \beta, \delta)$ [10], if its probability density function (pdf) is given by*

$$f(x) = c[(x - \beta)^2 + \delta]\phi(x; \mu, \sigma^2), \quad x \in \mathbb{R}, \tag{1}$$

*where $\mu, \beta \in \mathbb{R}$ are location parameters, $\sigma > 0$ and $\delta \geq 0$ denote respectively the scale and bimodality parameters, $\phi(\cdot; \mu, \sigma^2)$ is the normal pdf with location $\mu$ and scale $\sigma$ parameters; and $c = [\lambda^2 + \sigma^2 + \delta]^{-1}$, with $\lambda = \beta - \mu$.*

The mean and variance of $X$ are given by

$$E[X] = \mu - 2c\lambda\sigma^2, \tag{2}$$
$$Var[X] = c^2\sigma^2(3\sigma^4 + 4\delta\sigma^2 + [\lambda^2 + \delta]^2), \tag{3}$$

respectively. Equation (1) shows that $X \sim N(\mu, \sigma^2)$ as $\delta \to \infty$ or $|\beta| \to \infty$. The pdf of Equation (1) can also be expressed in a standardized form as is presented next.

**Definition 2.** *A random variable Y has a BSSN distribution, with location parameters $\mu, \beta \in \mathbb{R}$, positive scale parameter $\sigma$, and non-negative bimodality parameter $\delta$ if its pdf is*

$$f(y) = \left[c\sigma y^2 - 2c\lambda y + \frac{1}{\sigma} - 1\right]\phi(y), \quad y \in \mathbb{R},$$

*where $y = (x - \mu)/\sigma$, $\phi(\cdot)$ is the standardized normal pdf with location 0 and scale 1, and c and $\lambda$ are defined as in Equation (1).*

Figure 1 portrays various plots of the BSSN pdf, accommodating various shapes in terms of skewness, kurtosis and bimodality. We observed that bimodality is presented for smallest $\delta$, see also Proposition 2.4 in [10]. In addition, the $\mu$ and $\beta$ parameters allows accomodating skewness and kurtosis.

For a random sample $\mathbf{X} = (X_1, \ldots, X_n)^\top$ with pdf given in Equation (1), the log-likelihood function can be written as

$$\ell(\boldsymbol{\theta}; \mathbf{X}) = n \log c + \sum_{m=1}^{n} \log[(X_m - \beta)^2 + \delta] - \frac{1}{2} \sum_{m=1}^{n} Y_m^2, \tag{4}$$

where $Y_m = (X_m - \mu)/\sigma$, $m = 1, \ldots, n$, and $\boldsymbol{\theta} = (\mu, \sigma^2, \beta, \delta)^\top$. Therefore, the MLE $\widehat{\boldsymbol{\theta}}$ is obtained by maximizing the function (4). The Fisher Information Matrix (FIM) related to maximum likelihood equations and derivatives with respect to $\boldsymbol{\theta}$, is

$$\mathbf{I}(\boldsymbol{\theta}) = \begin{pmatrix} I_{\mu\mu} & I_{\sigma\mu} & I_{\beta\mu} & I_{\delta\mu} \\ I_{\mu\sigma} & I_{\sigma\sigma} & I_{\beta\sigma} & I_{\delta\sigma} \\ I_{\mu\beta} & I_{\sigma\beta} & I_{\beta\beta} & I_{\delta\beta} \\ I_{\mu\delta} & I_{\sigma\delta} & I_{\beta\delta} & I_{\delta\delta} \end{pmatrix}, \tag{5}$$

where its elements denoted by $I_{\theta_i \theta_j} = -E[\partial^2 \ell(\boldsymbol{\theta}; \mathbf{X})/\partial \theta_i \partial \theta_j]$, $\theta_k = \{\mu, \sigma^2, \beta, \delta\}$, $k = 1, 2, 3, 4$; are

$$I_{\mu\mu} = \frac{2nc}{\sigma^4} - n\left(\frac{2c\lambda}{\sigma^4}\right)^2 + \frac{n}{\sigma^2},$$

$$I_{\sigma\sigma} = 2n\sigma^2 - 4nc^2\left(\frac{\delta + \lambda^2}{\sigma^2}\right)^2,$$

$$I_{\beta\beta} = I_{\mu\mu} - \frac{n}{\sigma^2} - 2\sum_{i=1}^{n} \frac{\delta - (X_i - \beta)^2}{[\delta + (X_i - \beta)^2]^2},$$

$$I_{\delta\delta} = -n\left(\frac{c}{\sigma^4}\right)^2 + \sum_{i=1}^{n} \frac{1}{[\delta + (X_i - \beta)^2]^2},$$

$$I_{\mu\sigma} = -4n\lambda\left(\frac{c}{\sigma^2}\right)^2 - 2n(\overline{X} - \mu) = I_{\sigma\mu},$$

$$I_{\mu\beta} = \frac{n}{\sigma^2} - I_{\mu\mu} = I_{\beta\mu},$$

$$I_{\mu\delta} = 2n\lambda\left(\frac{c}{\sigma^4}\right)^2 = I_{\delta\mu},$$

$$I_{\sigma\beta} = 4n\lambda\left(\frac{c}{\sigma^2}\right)^2 = I_{\beta\sigma},$$

$$I_{\sigma\delta} = \frac{I_{\sigma\beta}}{2} = I_{\delta\sigma},$$

$$I_{\beta\delta} = -I_{\mu\delta} - 2\sum_{i=1}^{n} \frac{X_i - \beta}{[\delta + (X_i - \beta)^2]^2} = I_{\delta\beta},$$

where $\overline{X} = \frac{1}{n}\sum_{i=1}^{n} X_i$. It can be seen that FIM of Equation (5) is regular, for all $\delta \geq 0$.

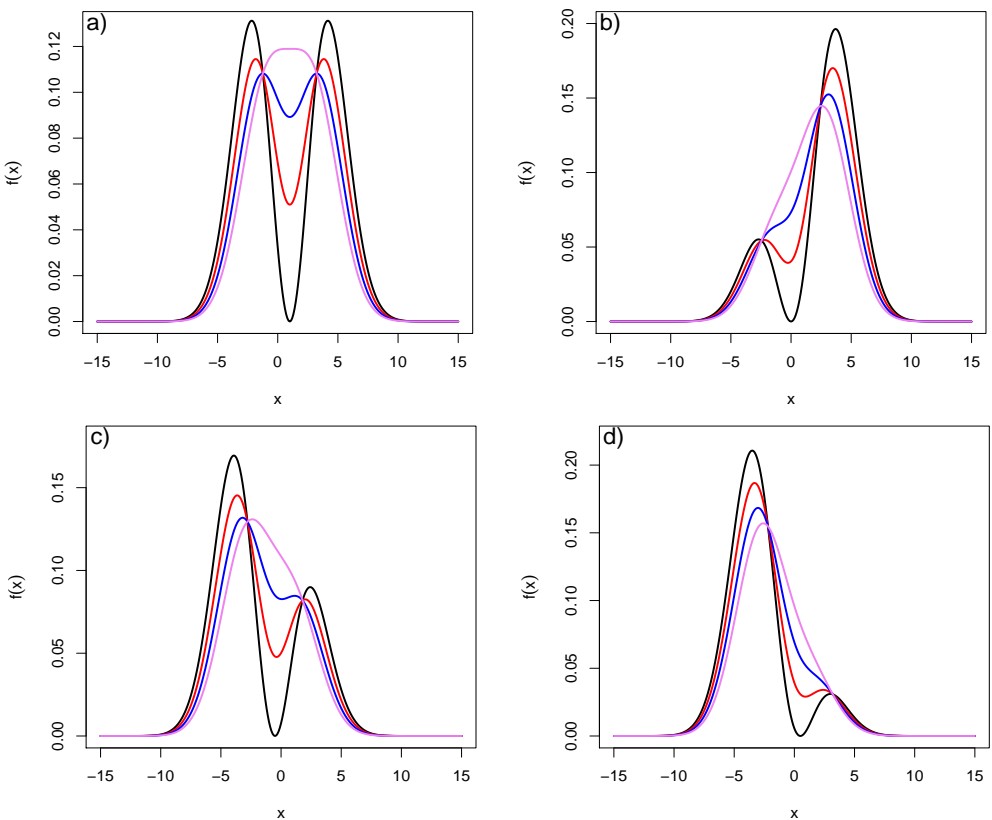

**Figure 1.** Various shapes of the pdfs of $X \sim BSSN(\mu, \sigma^2, \beta, \delta)$, with $\sigma^2 = 5$, $\delta = 0, 2, 5, 10$ (black, red, blue and violet lines, respectively); and (**a**) $\mu = 1, \beta = 1$, (**b**) $\mu = 1, \beta = 0$, (**c**) $\mu = -1, \beta = -0.5$, and (**d**) $\mu = -1, \beta = 0.5$ parameters.

## 3. Information Measures

In the next subsections, we present the main results of information measures for BSSN distribution.

### 3.1. Shannon Entropy

The entropy concept is attributed to uncertainty of information or mathematical contrariety of information. Of all possible entropies presented in the literature, we focus on Shannon entropy (SE) [13]. The SE of a random variable $Z$ with pdf $f(z)$ is defined as the expected value given by

$$\mathcal{H}(Z) = -E[\log f(Z)] = -\int_{\mathbb{R}} f(z) \log f(z) dz, \tag{6}$$

where $E[g(Z)]$ denotes the *expected information* in $Z$ for a function $g(z)$. In this case, SE is the expected value of the function $g(z) = -\log f(z)$, which satisfies $g(1) = 0$ and $g(0) = \infty$. We extend this notation in all expected values expressed in this paper.

**Proposition 1.** *Let* $X \sim BSSN(\mu, \sigma^2, \beta, \delta)$ *with pdf defined in Equation* (1)*, the SE of $X$ is given by*

$$\mathcal{H}(X) = \frac{1}{2} \log \left( \frac{2\pi\sigma^2}{c^2} \right) + \frac{(c\sigma)^2}{2} \left[ 3\sigma^2 + 4\delta - 4\lambda^2 + \left( \frac{\lambda^2 + \delta}{\sigma} \right)^2 \right] - E \left[ \log\{(X - \beta)^2 + \delta\} \right], \tag{7}$$

*with* $\lambda = \beta - \mu$.

**Proof.** From Equation (1), we have

$$\log f(x) = \log c + \log\left[ (x - \beta)^2 + \delta \right] - \frac{1}{2} \log(2\pi\sigma^2) - \frac{1}{2\sigma^2}(x - \mu)^2.$$

Then, from the definition of SE given in Equation (6), we have

$$\begin{aligned}
\mathcal{H}(X) &= -\int_{\mathbb{R}} f(x) \log c\, dx + \int_{\mathbb{R}} \frac{1}{2} \log(2\pi\sigma^2) f(x) dx \\
&\quad - \int_{\mathbb{R}} \log\left[ (x - \beta)^2 + \delta \right] f(x) dx + \int_{\mathbb{R}} \frac{1}{2\sigma^2}(x - \mu)^2 f(x) dx, \\
&= -\log c + \frac{1}{2} \log(2\pi\sigma^2) + \frac{1}{2\sigma^2} E[(x - \mu)^2] - E \left[ \log\left\{ (X - \beta)^2 + \delta \right\} \right]. \tag{8}
\end{aligned}$$

Given that $E[(X - \mu)^2] = Var[X - \mu] + E[X - \mu]^2 = Var[X] + (E[X] - \mu)^2$, the result for $\mathcal{H}(X)$ yields from Equations (2) and (3) and some basic algebra. □

For any $\delta$, the expected values of Equation (7) are not directly computable. However, the integrals are evaluated numerically using the `integrate` function of `R` software's [15] QUADPACK routine [16]. Several cases of SE given in Equation (7) are illustrated in the left panel of Figure 2 for $\delta = 0.1$ to 20. SE is positive and reaches its maximum value for largest values of $\beta$ and $0 < \delta < 5$ (where more bimodality exists). As is highlighted in Section 2, the SE of BSSN random variable tends to SE of a normal one,

$$\mathcal{H}(X) = \frac{1}{2} \log(2\pi\sigma^2 e) = \mathcal{H}(X_N),$$

as $\delta \to \infty$ or $|\beta| \to \infty$, where $X_N \sim N(\mu, \sigma^2)$ [17]. Therefore, for highest values of $\delta$, the SE decreases and converges to normal SE, $H(X_N) = 2.224$, with $\sigma^2 = 5$. It can be shown that $H(X) = \mathcal{H}(X_N)$ for $\delta \approx 500$.

From the expected value given in Equation (8), we could consider the polynomial of second order, $p(x) = x^2 - 2x\beta + \beta^2 + \delta$. This polynomial has determinant given by $\Delta = -4\delta$. Given that $\delta \geq 0$, we have two cases for possible roots, $u_1$ and $u_2$, of $p(x) = (x - u_1)(x - u_2)$:

(i) $\delta = 0 \Rightarrow \Delta = 0$: $u_1 = u_2 = \beta$ (real and equal roots). Thus, $f(x) = c^*(x - \beta)^2 \phi(x; \mu, \sigma^2)$, with $c^* = [\lambda^2 + \sigma^2]^{-1}$. However, for this case $X$ does not present bimodality, so $p(x) \neq 0$ for all $x \in \mathbb{R} \setminus \{\beta\}$.

(ii) $\delta > 0 \Rightarrow \Delta < 0$: $u_1 = \beta + i\sqrt{\delta}$ and $u_2 = \beta - i\sqrt{\delta}$, $i = \sqrt{-1}$ (complex and different roots). However, $x$ is defined in the real line, $\mathbb{R}$.

Considering cases (i) and (ii), the SE exists and is finite if $p(x) \neq 0$ for all $x \in \mathbb{R}/\{\beta\}$, $\delta = 0$, and for all $x \in \mathbb{R}$, $\delta > 0$. These cases are illustrated in the right panel of Figure 2. Red dots are related to roots without real part ($\beta = 0$) and the other dots are related to $\beta \neq 0$. Given that $\Delta < 0$, several dots are related to $\delta$, shaped like circles.

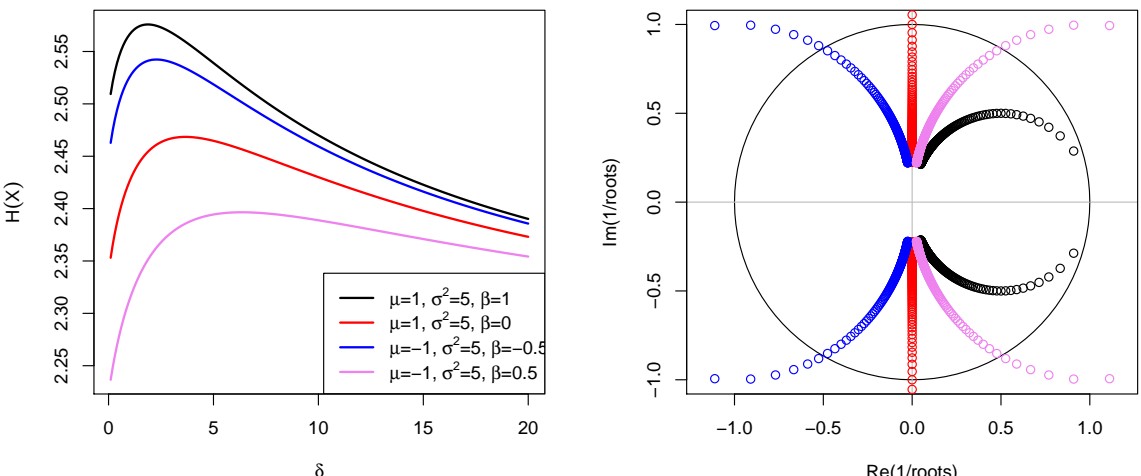

**Figure 2.** (**Left**) Shannon entropy for $X \sim BSSN(\mu, \sigma^2, \beta, \delta)$ using several combinations of $\mu$, $\beta$ and $\delta = 0.1, 0.2, \ldots, 20$. (**Right**) Inverse roots of $p(x) = (x - u_1)(x - u_2)$ in the unit circle for the same values of $\beta$ and $\delta$ used in the left panel, where the inverse roots, $1/u_1$ and $1/u_2$ are plotted in their real (x axis) and imaginary (y axis) parts, respectively.

### 3.2. Kullback-Leibler Divergence

Another measure related to the SE is the Kullback-Leibler (KL, [14]) divergence. It measures the degree of divergence between the distributions of two random variables, $Z_1$ and $Z_2$, with pdf $f(z_1)$ and $g(z_2)$, respectively. The KL divergence of the pdf of $Z_1$ from the pdf of $Z_2$ is defined by

$$\mathcal{K}(Z_1, Z_2) = E\left[\log\left\{\frac{f(z)}{g(z)}\right\}\right] = \int_{\mathbb{R}} f(z) \log\left\{\frac{f(z)}{g(z)}\right\} dz, \tag{9}$$

where, as indicated in the notation, the expectation is defined with respect to the pdf $f(z_1)$. We note that $\mathcal{K}(Z_j, Z_j) = 0$, $j = 1, 2$, but again $\mathcal{K}(Z_j, Z_k) \neq \mathcal{K}(Z_k, Z_j)$, $j, k = 1, 2$, $j \neq k$, at least that $Z_j \overset{d}{=} Z_k$, i.e., the KL divergence is not symmetric. An important property of KL divergence is that is non-negative: $\mathcal{K}(Z_j, Z_k) \geq 0$, $j, k = 1, 2$, $j \neq k$, for all $Z_1, Z_2$. Given that KL divergence does not satisfy the triangular inequality, it must be interpreted as a pseudo-distance measure [17].

**Proposition 2.** *Let $Z_j \sim BSSN(\mu_j, \sigma_j^2, \beta_j, \delta_j)$, $j = 1, 2$, both with pdf defined in Equation (1), the KL divergence between $Z_1$ and $Z_2$ is given by*

$$\begin{aligned}
\mathcal{K}(Z_1, Z_2) &= \log\left(\frac{c_1\sigma_2}{c_2\sigma_1}\right) + \frac{1}{2}\left(\frac{1}{\sigma_2^2} - \frac{1}{\sigma_1^2}\right) c_1^2\sigma_1^2(3\sigma_1^4 + 4\delta_1\sigma_1^2 + [\lambda_1^2 + \delta_1]^2) \\
&\quad + \frac{1}{2\sigma_2^2}(\mu_1 - \mu_2 - 2c_1\lambda_1\sigma_1^2)^2 - 2c_1^2\lambda_1^2\sigma_1^2 + E\left[\log\left\{\frac{(Z_1 - \beta_1)^2 + \delta_1}{(Z_1 - \beta_2)^2 + \delta_2}\right\}\right],
\end{aligned} \tag{10}$$

*with* $c_j = [\lambda_j^2 + \sigma_j^2 + \delta_j]^{-1}$, $\lambda_j = \beta_j - \mu_j$, $j = 1, 2$.

**Proof.** Assuming that $Z_1$ and $Z_2$ have pdf $f$ and $g$, respectively; from Equation (1) we get

$$\log g(x) = \log c_2 + \log\left[(x - \beta_2)^2 + \delta_2\right] - \frac{1}{2}\log(2\pi\sigma_2^2) - \frac{1}{2\sigma_2^2}(x - \mu_2)^2.$$

Then, from definition of KL divergence given in Equation (9), we get

$$
\begin{aligned}
\mathcal{K}(Z_1, Z_2) &= -\int_{\mathbb{R}} f(z_1)\log g(z_1)dx - H(Z_1) \\
&= -\int_{\mathbb{R}} \log\left[(z_1 - \beta_2)^2 + \delta_2\right]f(z_1)dz_1 + \int_{\mathbb{R}} \frac{1}{2\sigma_2^2}(z_1 - \mu_2)^2 f(z_1)dz_1 \\
&\quad - \int_{\mathbb{R}} f(z_1)\log c_2 dz_1 + \int_{\mathbb{R}} \frac{1}{2}\log(2\pi\sigma_2^2)f(z_1)dz_1 - H(Z_1) \\
&= \frac{1}{2}\log\left(\frac{2\pi\sigma_2^2}{c_2^2}\right) + \frac{1}{2\sigma_2^2}E[(Z_1 - \mu_2)^2] - E\left[\log\left\{(Z_1 - \beta_2)^2 + \delta_2\right\}\right] - H(Z_1) \\
&= \frac{1}{2}\log\left(\frac{2\pi\sigma_2^2}{c_2^2}\right) + \frac{1}{2\sigma_2^2}\left[Var[Z_1] + (E[Z_1] - \mu_2)^2\right] \\
&\quad - \frac{1}{2}\log\left(\frac{2\pi\sigma_1^2}{c_1^2}\right) - \frac{1}{2\sigma_1^2}\left[Var[Z_1] + (E[Z_1] - \mu_1)^2\right] \\
&\quad + E\left[\log\{(Z_1 - \beta_1)^2 + \delta_1\}\right] - E\left[\log\left\{(Z_1 - \beta_2)^2 + \delta_2\right\}\right].
\end{aligned}
\tag{11}
$$

Given that $E[(Z_1 - \mu_2)^2] = Var[Z_1 - \mu_2] + E[Z_1 - \mu_2]^2 = Var[Z_1] + (E[Z_1] - \mu_2)^2$, the result yields from Equations (2) and (3), Proposition 1, and some basic algebra. □

For any $\delta_j$, $j = 1, 2$, the expected values of Equation (10) are not directly computable. However, the integrals were evaluated numerically using the `integrate` function of QUADPACK routine [16]. Besides, we are considering two polynomials of second order, $p_j(x) = x^2 - 2x\beta_j + \beta_j^2 + \delta_j$, with determinants given by $\Delta_j = -4\delta_j$, $j = 1, 2$, respectively. Given that $\delta_j \geq 0$, we get four cases for possible roots, $u_{j,k}$ of $p_j(x) = (x - u_{j,k})(x - u_{j,k})$, $j, k = 1, 2$:

(i) $\delta_j = 0 \Rightarrow \Delta_j = 0$, $j = 1, 2$: $u_{1,1} = u_{1,2} = \beta_1$ and $u_{2,1} = u_{2,2} = \beta_2$ (real and equal roots). Thus, $f(x) = c_1(x - \beta_1)^2\phi(x; \mu_1, \sigma_1^2)$, with $c_1 = [\lambda_1^2 + \sigma_1^2]^{-1}$, and $g(x) = c_2(x - \beta_2)^2\phi(x; \mu_2, \sigma_2^2)$, with $c_2 = [\lambda_2^2 + \sigma_2^2]^{-1}$. However, neither densities presents bimodality. Thus, $p_1(x) \neq 0$, for all $x \in \mathbb{R} \setminus \{\beta_1\}$, and $p_2(x) \neq 0$, for all $x \in \mathbb{R} \setminus \{\beta_2\}$.

(ii) $\delta_j > 0 \Rightarrow \Delta_j < 0$, $j = 1, 2$: $u_{j,1} = \beta_j + i\sqrt{\delta_j}$ and $u_{j,2} = \beta_j - i\sqrt{\delta_j}$ (complex and different roots). However, $z_1$ is defined in the real line, $\mathbb{R}$.

(iii) $\delta_1 = 0 \Rightarrow \Delta_1 = 0$, $\delta_2 > 0 \Rightarrow \Delta_2 < 0$: $u_{1,1} = u_{1,2} = \beta_1$, $u_{2,1} = \beta_2 + i\sqrt{\delta_2}$ and $u_{2,2} = \beta_2 - i\sqrt{\delta_2}$ (complex and different roots). Thus, $f(x) = c_1(x - \beta_1)^2\phi(x; \mu_1, \sigma_1^2)$, with $c_1 = [\lambda_1^2 + \sigma_1^2]^{-1}$. However, $f(x)$ does not present bimodality and $z_1$ is defined in the real line, $\mathbb{R}$. So, $p_1(x) \neq 0$, for all $x \in \mathbb{R} \setminus \{\beta_1\}$.

(iv) $\delta_1 > 0 \Rightarrow \Delta_1 < 0$, $\delta_2 = 0 \Rightarrow \Delta_2 = 0$, : $u_{2,1} = u_{2,2} = \beta_2$, $u_{1,1} = \beta_1 + i\sqrt{\delta_1}$ and $u_{1,2} = \beta_1 - i\sqrt{\delta_1}$ (complex and different roots). Hence, $g(x) = c_2(x - \beta_2)^2\phi(x; \mu_2, \sigma_2^2)$, with $c_2 = [\lambda_2^2 + \sigma_2^2]^{-1}$. However, $g(x)$ does not present bimodality and $z_1$ is defined in the real line, $\mathbb{R}$. Therefore, $p_2(x) \neq 0$, for all $x \in \mathbb{R} \setminus \{\beta_2\}$.

All of these cases are analogous to those illustrated in the right panel of Figure 2.

**Corollary 1.** *Let $Z \sim BSSN(\mu, \sigma^2, \beta, \delta)$ and $Z_0 \sim BSSN(\mu, \sigma^2, \beta, \delta_0)$, both with pdf defined in Equation (1), the KL divergence between $Z$ and $Z_0$ is given by*

$$\mathcal{K}(Z, Z_0) = \log\left(\frac{\lambda^2 + \sigma^2 + \delta_0}{\lambda^2 + \sigma^2 + \delta}\right) + E\left[\log\left\{\frac{(Z-\beta)^2 + \delta}{(Z-\beta)^2 + \delta_0}\right\}\right], \tag{12}$$

*with $\lambda = \beta - \mu$.*

**Proof.** The result is straightforward from Proposition 2 (by replacing $\mu = \mu_1 = \mu_2$, $\sigma = \sigma_1 = \sigma_2$, $\beta = \beta_1 = \beta_2$, $\delta = \delta_1$ and $\delta_2 = \delta_0$) and some basic algebra.  □

As is highlighted in Section 2, the KL divergence between two BSSN random variables tends to a KL divergence between two normal ones [17],

$$\mathcal{K}(Z_1, Z_2) = \frac{1}{2}\left\{\log\left(\frac{\sigma_2^2}{\sigma_1^2}\right) + \frac{\sigma_1^2}{\sigma_2^2} + \frac{(\mu_1 - \mu_2)^2}{\sigma_2^2} - 1\right\} = \mathcal{K}(X_1, X_2),$$

as $\delta_j \to \infty$ or $|\beta_j| \to \infty$, where $X_j \sim N(\mu_j, \sigma_j^2)$, $j = 1, 2$.

Figure 3 (left) illustrates the numerical behavior of the KL divergence between two BSSN distributions under different $\delta_1$ and $\delta_2$ parameters. Specifically, we can observe from there the behavior of the KL divergence given in Proposition 2, where for $\delta_1 \approx \delta_2$, the KL divergence tends to zero but is always non-negative. For $\delta_1 \neq \delta_2$, we observe that KL divergence has the highest values. The right panel illustrates the cases $\delta_1 = \{0.5, \ldots, 100\}$ and $\delta_2 = \{0, 2, 5, 10\}$, where the KL divergence converges to 1.269 when $\delta_2 = 0$ (see Equation (12)), as $\delta_1 \to \infty$, and increases for $\delta_1$ between 0 and 100. For $\delta_1, \delta_2 > 0$, the KL divergence decreases because more similarity exists between bimodality parameters.

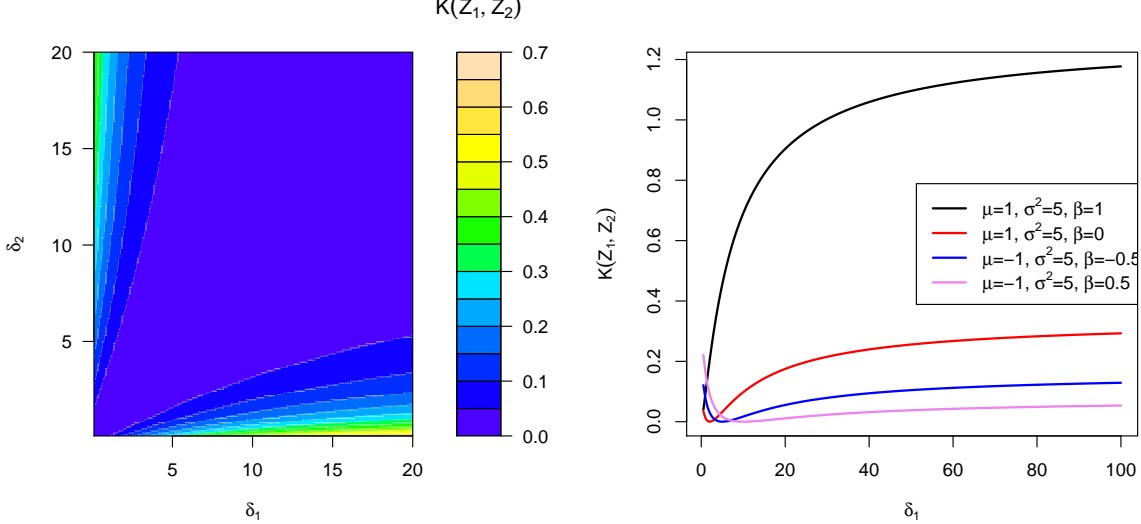

**Figure 3.** (**Left**) KL divergence between $Z_1 \sim BSSN(1, 5, 1, \delta_1)$ and $Z_2 \sim BSSN(1, 5, 1, \delta_2)$, for $\delta = 0.1, 0.2, \ldots, 20$. (**Right**) KL divergence between $Z_1$ and $Z_2$, $Z_j \sim BSSN(\mu, \sigma^2, \beta, \delta_j)$, $j = 1, 2$, for $\delta_1 = 0.5, \ldots, 100$, $\delta_2 = 0, 2, 5, 10$, and the same parameters $\mu$, $\sigma^2$ and $\beta$ of Figures 1 and 2.

### 3.3. Jeffreys Divergence

As KL divergence is not symmetric, Jeffrey's (J) divergence [18] is considered as a symmetric version of the KL divergence, which is defined as

$$\mathcal{J}(Z_1, Z_2) = \mathcal{K}(Z_1, Z_2) + \mathcal{K}(Z_2, Z_1) = \mathcal{J}(Z_2, Z_1). \tag{13}$$

The J divergence does not satisfy the triangular inequality of distance, so it is a pseudo-distance measure.

**Corollary 2.** *Let $Z_j \sim BSSN(\mu_j, \sigma_j^2, \beta_j, \delta_j)$, $j = 1, 2$, both with pdf defined in Equation (1), the J divergence between $Z_1$ and $Z_2$ is given by*

$$
\begin{aligned}
\mathcal{J}(Z_1, Z_2) &= \frac{1}{2}\left(\frac{1}{\sigma_2^2} - \frac{1}{\sigma_1^2}\right)\left[c_1^2\sigma_1^2(3\sigma_1^4 + 4\delta_1\sigma_1^2 + [\lambda_1^2 + \delta_1]^2) - c_2^2\sigma_2^2(3\sigma_2^4 + 4\delta_2\sigma_2^2 + [\lambda_2^2 + \delta_2]^2)\right] \\
&+ \frac{1}{2\sigma_2^2}(\mu_1 - \mu_2 - 2c_1\lambda_1\sigma_1^2)^2 + \frac{1}{2\sigma_1^2}(\mu_2 - \mu_1 - 2c_2\lambda_2\sigma_2^2)^2 - 2c_1^2\lambda_1^2\sigma_1^2 - 2c_2^2\lambda_2^2\sigma_2^2 \\
&+ E\left[\log\left\{\frac{(Z_1 - \beta_1)^2 + \delta_1}{(Z_1 - \beta_2)^2 + \delta_2}\right\}\right] + E\left[\log\left\{\frac{(Z_2 - \beta_2)^2 + \delta_2}{(Z_2 - \beta_1)^2 + \delta_1}\right\}\right].
\end{aligned} \tag{14}
$$

*with $c_j = [\lambda_j^2 + \sigma_j^2 + \delta_j]^{-1}$, $\lambda_j = \beta_j - \mu_j$, $j = 1, 2$.*

**Proof.** The result is straightforward from the definition given in Equation (13), Proposition 2, and some basic algebra. ☐

As mentioned in Section 2, the J divergence between two BSSN random variables tends to a J divergence between two normal ones [17],

$$
\mathcal{J}(Z_1, Z_2) = \frac{1}{2}\left\{\frac{\sigma_1^2}{\sigma_2^2} + (\mu_1 - \mu_2)^2\left(\frac{1}{\sigma_2^2} + \frac{1}{\sigma_1^2}\right) - 2\right\} = \mathcal{J}(X_1, X_2),
$$

as $\delta_j \to \infty$ or $|\beta_j| \to \infty$, where $X_j \sim N(\mu_j, \sigma_j^2)$, $j = 1, 2$.

## 4. Bimodality Test

First, an analytical tool is necessary to determine a set of values for $\delta$ where bimodality exists. Following Proposition 2.5 of [10], the steps presented next determine these values for given $\mu$, $\beta$ and $\sigma^2$ parameters.

### 4.1. Bimodality

Let $f^{(k)}(x) = \frac{\partial^k f(x)}{\partial x^k}$ be the $k$th derivative of $f(x)$ with respect to $x$, $k = 1, 2$, we have

$$
\begin{aligned}
f^{(1)}(x) &= 2c\left\{(x - \beta) - \frac{1}{2\sigma^2}(x - \mu)[(x - \beta)^2 + \delta]\right\}\phi(x; \mu, \sigma^2), \\
f^{(2)}(x) &= 2c\left\{1 + \frac{1}{2\sigma^4}(x - \mu)^2[(x - \beta)^2 + \delta] - \frac{1}{2\sigma^2}[(x - \beta)^2 + 4(x - \mu)(x - \beta) + \delta]\right\}\phi(x; \mu, \sigma^2).
\end{aligned}
$$

Thus, the pdf of Equation (1) is bimodal if a $\delta_0 \geq 0$ like $\delta < \delta_0$ exists for the following cases:

(i)  if $\mu = \beta$, thus $\delta_0 = 2\sigma^2$;

(ii) if $\mu \neq \beta$, thus $f^{(1)}(x) = 0$ implies to find three roots, $v_1$, $v_2$ and $v_3$ ($v_1 < v_2 < v_3$), of the polynomial of degree three, $r(x) = a_3x^3 + a_2x^2 + a_1x + a_0 = 0$, with

$$
\begin{aligned}
a_3 &= \frac{1}{2\sigma^2}, \\
a_2 &= -\frac{1}{2\sigma^2}(2\beta + \mu), \\
a_1 &= \frac{1}{2\sigma^2}(\beta^2 + 2\beta\mu + \delta - 2\sigma^2), \\
a_0 &= -\frac{1}{2\sigma^2}(\beta^2\mu + \delta\mu - 2\beta\sigma^2).
\end{aligned}
$$

For given $\mu$, $\sigma^2$ and $\beta$ parameters, $\mu \neq \beta$, the polynomial $r(x)$ can be solved for $v_2$ in terms of $\delta$ and inequality $f^{(2)}(v_2) > 0$ can be used to determine $\delta_0$. This implies that

$$
\delta < \frac{2\sigma^4 + (v_2 - \mu)^2(v_2 - \beta)^2 - \sigma^2(v_2 - \beta)(5v_2 - 4\mu - \beta)}{\sigma^2 - (v_2 - \mu)^2} = \delta_0. \tag{15}
$$

Therefore, since $\delta < \delta_0$, the upper bound given in Equation (15) can be used for detecting bimodality if $\delta_0 > 0$ for a given root $v_2$ of $r(x)$, $v_1 < v_2 < v_3$, and $\mu$, $\sigma^2$ and $\beta$ parameters.

### 4.2. Asymptotic Test

The results given in [12] can be applied, for example, to construct a bimodality test from the KL divergence presented in Corollary 1 between a regular BSSN distribution and a BSSN distribution without bimodality. Specifically, consider a random sample $X_1, \ldots, X_n$ from $X \sim BSSN(\mu, \sigma^2, \beta, \delta)$ and the null ($H_0$) and alternative ($H_1$) hypothesis

$$
H_0 : \delta \leq \delta_0 \quad \text{versus} \quad H_1 : \delta > \delta_0, \tag{16}
$$

where the null and alternative hypothesis refers to bimodality and unimodality, respectively. Thus the BSSN random variable $X$ becomes a $BSSN(\mu, \sigma^2, \beta, \delta_0)$ random variable for a specific value $\delta_0$ under $H_0$. The $\delta_0$ could be selected using, for example, the criteria explained in cases (i) and (ii) of Section 4.1.

**Proposition 3.** *Let* $\widehat{\boldsymbol{\theta}} = (\widehat{\mu}, \widehat{\sigma^2}, \widehat{\beta}, \widehat{\delta})^\top$ *be the MLE of* $\boldsymbol{\theta} = (\mu, \sigma^2, \beta, \delta)^\top$ *as in Section 2, and* $\widehat{\boldsymbol{\theta}}_0 = (\widehat{\mu}, \widehat{\sigma^2}, \widehat{\beta}, \delta_0)^\top$. *Therefore, under* $H_0$ *we have*

$$
\mathcal{S}_{\mathcal{K}}(\widehat{\boldsymbol{\theta}}, \widehat{\boldsymbol{\theta}}_0) = 2n\widehat{\mathcal{K}}(Z, Z_0) \xrightarrow[n \to \infty]{d} \chi_1^2, \tag{17}
$$

*where* $\chi_1^2$ *denotes the chi-square distribution with 1 degree of freedom, and* $\widehat{\mathcal{K}}(Z, Z_0)$ *is the MLE of* $\mathcal{K}(Z, Z_0)$ *defined in Equation (12) of Corollary 1.*

**Proof.** The result is straightforward from ([12], p. 375). □

Under specifications of Proposition 3, the statistic $\mathcal{S}_{\mathcal{K}}(\widehat{\boldsymbol{\theta}}, \widehat{\boldsymbol{\theta}}_0)$ depends only on $\widehat{\delta}$ and $n$. As stated in Sections 2 and 3, unimodality is typically obtained from the BSSN class at $\delta \leq \delta_0$. Given that FIM is regular and regularity conditions (i), (ii), and (iii) stated in ([12], p. 375) are satisfied, it is possible to test bimodality via hypothesis testing of Proposition 3. Let

$$
\mathcal{C} = \{X_1, \ldots, X_n \mid \mathcal{S}_{\mathcal{K}}(\widehat{\boldsymbol{\theta}}, \widehat{\boldsymbol{\theta}}_0) \geq \chi_{1-\alpha}^2, \ 0 < \alpha < 1\}
$$

be the critical region related to (16), thus $\mathbb{P}(\chi_1^2 \leq \chi_{1-\alpha}^2) = 1 - \alpha$. Hence, from Proposition 3, evidence exists to accept the null hypothesis of bimodality given in Equation (16) at level $\alpha$ if

$$
\mathbb{P}(\chi_1^2 < \mathcal{S}_{\mathcal{K}}(\widehat{\boldsymbol{\theta}}, \widehat{\boldsymbol{\theta}}_0)) > 1 - \alpha \quad \text{or} \quad \mathbb{P}(\chi_1^2 > \mathcal{S}_{\mathcal{K}}(\widehat{\boldsymbol{\theta}}, \widehat{\boldsymbol{\theta}}_0)) \leq \alpha. \tag{18}
$$

The observed power of the asymptotic bimodality test can be obtained from Equations (17) and (18), for different sample sizes and values of the bimodality parameter. These results were obtained from 1000 simulations for a nominal level of 5%. In each simulation, the estimation of the BSSN model's parameters was carried out by maximizing the likelihood function of Equation (4) over the parameter space $\theta$ and a random sample of size $n = 25, 50, 100$ and 200. To estimate the parameters and get their standard errors, first the random sample is obtained using the `rBSSN` function of `gamlssbssn` package [19]. Second, the log-likelihood function is computed using the pdf of Equation (1) implemented in the same package. Third,, the log-likelihood function is optimized using the `mle` function included in the `stats4` package of `R` software [15]. To avoid local maxima, the optimization routine was run using specific starting values used for random samples.

Table 1 shows that the proposed test is highly conservative since the observed rate of incorrect rejections of the bimodality hypothesis ($H_0$) is always lower than the nominal level, i.e., for $\delta \ll \delta_0$ and $\delta \gg \delta_0$, the observed power tends to increase and decrease, respectively. The proposed test is also more powerful in large samples ($n \geq 100$) and for $\delta > 0.5$. As expected, the power of the test increases with sample size, given that statistic $\mathcal{S}_{\mathcal{K}}(\widehat{\theta}, \widehat{\theta}')$ depends on $n$ although $\widehat{\mathcal{K}}(Z, Z_0)$ is small (Figure 3).

**Table 1.** Observed power (in %) of the proposed bimodality test using MLE of BSSN model from 1000 simulations for nominal level 5%, locations $\mu = 1$ and $\beta = 0$ (see Figure 1b), various values of bimodality parameters $\delta$ and $\delta_0$, and sample size $n$.

| | | $\delta_0$ | | | | | | |
|---|---|---|---|---|---|---|---|---|
| $n$ | $\delta$ | 0.5 | 1 | 2 | 3 | 5 | 7 | 10 |
| 25 | 0.5 | 25.40 | 17.63 | 19.78 | 34.31 | 59.97 | 75.69 | 86.49 |
| | 2 | 44.27 | 30.19 | 23.01 | 21.39 | 33.61 | 48.22 | 65.56 |
| | 5 | 75.19 | 56.85 | 34.51 | 26.77 | 25.05 | 31.91 | 38.73 |
| | 7 | 84.04 | 70.12 | 40.21 | 32.34 | 23.00 | 26.96 | 34.07 |
| 50 | 0.5 | 18.14 | 16.64 | 47.95 | 72.14 | 93.58 | 97.52 | 99.23 |
| | 2 | 62.58 | 35.97 | 23.89 | 33.05 | 59.77 | 77.14 | 87.29 |
| | 5 | 94.42 | 81.45 | 51.11 | 31.96 | 25.90 | 36.40 | 49.09 |
| | 7 | 97.68 | 87.83 | 64.03 | 47.04 | 26.01 | 26.72 | 36.61 |
| 100 | 0.5 | 19.70 | 29.65 | 81.74 | 95.53 | 99.75 | 99.87 | 100.00 |
| | 2 | 79.76 | 48.59 | 24.22 | 39.92 | 77.82 | 92.45 | 97.79 |
| | 5 | 99.90 | 96.20 | 69.70 | 43.40 | 26.03 | 36.60 | 59.50 |
| | 7 | 99.90 | 99.20 | 88.00 | 66.27 | 29.20 | 26.03 | 41.00 |
| 200 | 0.5 | 21.37 | 53.33 | 95.04 | 100.00 | 100.00 | 100.00 | 100.00 |
| | 2 | 95.80 | 70.10 | 24.92 | 51.30 | 93.20 | 99.30 | 100.00 |
| | 5 | 100.00 | 100.00 | 93.20 | 60.10 | 23.30 | 45.50 | 80.10 |
| | 7 | 100.00 | 100.00 | 99.40 | 88.90 | 38.80 | 24.20 | 45.60 |

## 5. Application to Sea Surface Temperature Data

A real application in this section illustrates the performance of the asymptotic bimodality test. Specifically, we considered the Sea Surface Temperature (SST) data sets presented in [20], which were recorded from 2012 to 2014 by scientific observers of the northern Chilean longline fleet (industrial and artisanal, 21°31′–36°39′ LS and 71°08′–85°52′ LW). Contreras-Reyes et al. [20] proposed the Skewed Reflected Gompertz (SRG) model based on two-piece distributions [21] as suitable for interpreting annual bimodal and asymmetric SST data. The SRG distribution produces two-piece asymmetric and bimodality behavior of Gompertz (GZ) density.

To estimate the parameters and get their standard errors, the log-likelihood function and its optimization were carried out (see Section 4.2). However, to avoid local maxima, the optimization routine was run using specific starting values obtained by visual inspection of histograms, which are widely scattered in the parameter space. To evaluate the goodness of fit test, the Kolmogorov–Smirnov (K–S), Anderson–Darling (A–D), and Cramer-von Mises (C–V) tests were considered for all

models. These are commonly used to analyze the goodness of fit test of a particular distribution (see e.g., [20,21]). The test are implemented with the `goftest` package [22] of R software, and all considered the cumulative distribution function pBSSN of `gamlssbssn` package [19]. The proposed asymptotic bimodality test is compared with a nonparametric approach-based asymptotic test (DIPtest), implemented in the `diptest` package/function [23].

Considering the smallest Akaike (AIC) and Schwarz (BIC) information criteria values, we observed in Table 2 that BSSN performs better than the SRG model (and the other competitors, see AIC and BIC values reported in Table 1 of [20]). In addition, considering the K–S, A–D, and C–V test for a 95% confidence level, BSSN fits perform well for all years (*p*-values higher than 0.05 mean appropriate goodness of fit). Figure 4 illustrates this performance, where more than one mode is presented in histograms. The most notorious bimodality emerged for 2014.

Parameters estimated from the BSSN model, presented in Table 2, are used to perform the SE and KL divergence for SST in each year and for the asymptotic test of Section 4.2. The determination of $\delta_0$ was conducted using the procedure explained in Section 4.1. The results of these analyses appear in Table 3, where $\widehat{K}(Z, Z_0)$ represents the KL divergence under null hypothesis. Shannon entropies illustrate that most SST information come from 2013. In addition, the asymptotic test presented in Table 3 is analogous for all years. In fact, the null hypothesis $H_0$ of bimodality is accepted at 95% confidence level according to Equation (18). This acceptance is reinforced by large sample size and by the DIPtest results, where rejection (*p*-value < 0.05) implies at least bimodality.

**Table 2.** Parameter estimates and their respective standard deviations (S.D) for SST by year based on BSSN model. For each fit, log-likelihood function $\ell(\theta)$ with $\theta = (\mu, \sigma^2, \beta, \delta)$, Akaike (AIC) and Schwarz (BIC) information criteria, and goodness of fit tests (Kolmogorov–Smirnov (K–S), Anderson–Darling (A–D), and Cramer–von Mises, (C–V)) are also reported with respective p-values in parenthesis.

| Year | Param. | Estim. | (S.D) | $\ell(\theta)$ | AIC | BIC | K–S | A–D | C–V |
|---|---|---|---|---|---|---|---|---|---|
| 2012 | $\mu$ | 19.007 | 0.078 | −1396.1 | 2800.3 | 2818.9 | 0.042 | 1.760 | 0.233 |
| ($n = 774$) | $\sigma^2$ | 1.434 | 0.020 | | | | (0.13) | (0.13) | (0.21) |
| | $\beta$ | 19.670 | 0.151 | | | | | | |
| | $\delta$ | 1.746 | 0.384 | | | | | | |
| 2013 | $\mu$ | 18.187 | 0.068 | −683.71 | 1375.4 | 1391.5 | 0.035 | 0.636 | 0.074 |
| ($n = 414$) | $\sigma^2$ | 0.886 | 0.044 | | | | (0.68) | (0.61) | (0.73) |
| | $\beta$ | 18.328 | 0.127 | | | | | | |
| | $\delta$ | 1.026 | 0.310 | | | | | | |
| 2014 | $\mu$ | 17.628 | 0.040 | −643.62 | 1295.2 | 1311.6 | 0.043 | 0.518 | 0.070 |
| ($n = 439$) | $\sigma^2$ | 0.550 | 0.054 | | | | (0.41) | (0.73) | (0.75) |
| | $\beta$ | 17.682 | 0.058 | | | | | | |
| | $\delta$ | 0.306 | 0.079 | | | | | | |

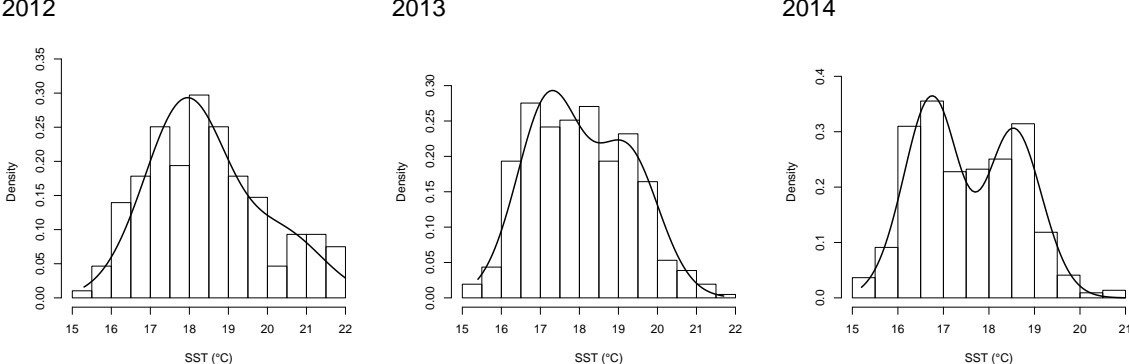

**Figure 4.** Histograms of SST data by year and their respective MLE fits of BSSN model (solid line).

**Table 3.** BSSN Shannon, $H(Z)$, KL divergence $\widehat{K}(Z, Z_0)$, statistic and respective p-values of Equation (17) are reported for SST data and for each year. All reported $H(Z)$, $\widehat{\delta}$, and $\widehat{K}(Z, Z_0)$ estimates considered the estimated parameters and sample size $n$ reported in Table 2.

| Method | Quantifier | 2012 | 2013 | 2014 |
|---|---|---|---|---|
| Proposed | $H(Z)$ | 1.613 | 1.634 | 1.455 |
| | $\widehat{\delta}$ | 1.746 | 1.026 | 0.306 |
| | $\delta_0$ | 9.273 | 2.534 | 2.579 |
| | $\widehat{K}(Z, Z_0)$ | 0.003 | 0.025 | 0.138 |
| | Statistic | 556.657 | 21.425 | 121.087 |
| | $p$-value | 0.971 | 0.999 | 1.000 |
| DIPtest | Statistic | 0.023 | 0.029 | 0.038 |
| | $p$-value | <0.01 | 0.016 | <0.01 |

## 6. Conclusions

We have presented a methodology to compute the Shannon entropy and the Kullback–Leibler and Jeffreys divergences for the family of bimodal skew-symmetric normal distributions. Given the regularity conditions accomplished by the BSSN distribution, specifically by the regularity of Fisher information matrix, an asymptotic test for bimodality was developed. A statistical application to South Pacific sea surface temperature was given. We illustrated that asymptotic tests in samples of three years were useful to detect strong evidence of bimodality. This approach can be applied to real models and used for data analysis in various systems, such as Artic Sea Temperature [24] and biological [25] data.

The main result is that information measures and asymptotic tests can be employed in bimodal distributions (if regularity conditions are accomplished [12]) and present enough flexibility in complex data. Compared with DIPtest [9,23] and kurtosis measure [7,8], the proposed asymptotic test for bimodality presented the following novelties: (i) it was built under a parametric approach (a known distribution); (ii) it was based on information measures; and (iii) it considered regularity conditions of BSSN distribution. In addition, the computation of information quantifiers of BSSN distributions is a more adequate tool, compared with information quantifiers obtained for finite mixture of flexible distributions, where Shannon entropy is approximated by bounds [26].

Finally, we encourage researchers to consider the proposed methodology for further investigations with other bimodal distributions, such as bimodal normal distribution [11], the extension proposed in Equation (19) of [10], or the generalized bimodal skew-normal distribution proposed by [27].

**Author Contributions:** J.E.C.-R. wrote the paper and contributed reagents/analysis/materials tools; conceived, designed and performed the experiments and analyzed the data. All authors have read and agreed to the published version of the manuscript.

**Acknowledgments:** I am grateful to Daniel Devia Cortés (IFOP) for providing access to the data used in this work. Author's research was fully supported by Grant FONDECYT (Chile) No. 11190116. The author thanks the editor and two anonymous referees for their helpful comments and suggestions. All R codes and data used in this paper are available by request to the corresponding author.

**Conflicts of Interest:** The author declares that there is no conflict of interest in the publication of this paper.

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
