# Peer review of "An Asymptotic Test for Bimodality Using The Kullback–Leibler Divergence"

_symmetry, doi:10.3390/sym12061013_

Round 1
Reviewer 1 Report
See file attached.

Author Response
Dear Reviewer:
We would like acknowledge this careful revision of our manuscript symmetry-821503: "An asymptotic test for bimodality using the Kullback-Leibler divergence". We are grateful that this manuscript can be considered for publication after mayor revision. We also thank the reviewer for all their valuable comments and constructive criticism. We have included (see attached file below), a detailed
point-by-point response to all the reviewer's comments and suggestions. The comments from the reviewer are listed in cursive italic letters, and updated lines appear in red in the manuscript.
Best regards,
Dr. Javier E. Contreras-Reyes

Reviewer 2 Report
This paper
looks at mathematical methods to test for bimodality in statistical data sets.
Particularly it can be sued to calculate the Shannon entropy and the KL divergences.
Such distributions arise across numerous fields
are common across a variety of fields and methods to better characterize bimodality will always be usefull.
A nice part of the paper is the authors than applied the method to a real model shich is the Sea Ice tempters showing
the usefulness of the measure. I think the paper will be valuable for data analysis on various systems and as
such could be published.

Author Response

(The authors gave the same response as above.)

Round 2
Reviewer 1 Report
The paper in its present form is suitable for publication.